# Cytokines and Immune Cells Profile in Different Tissues of Rodents Induced by Environmental Enrichment: Systematic Review

**DOI:** 10.3390/ijms231911986

**Published:** 2022-10-09

**Authors:** Matheus Santos de Sousa Fernandes, Gabriela Carvalho Jurema Santos, Tayrine Ordonio Filgueira, Dayane Aparecida Gomes, Elias Almeida Silva Barbosa, Tony Meireles dos Santos, Niels Olsen Saraiva Câmara, Angela Castoldi, Fabricio Oliveira Souto

**Affiliations:** 1Programa de Pós-Graduação em Neuropsiquiatria e Ciências do Comportamento, Centro de Ciências da Médicas, Universidade Federal de Pernambuco, Recife 50740-600, Brazil; 2Faculdade de Comunicação Turismo e Tecnologia de Olinda, Olinda 53030-010, Brazil; 3Programa de Pós-Graduação em Nutrição, Universidade Federal de Pernambuco, Recife 50740-600, Brazil; 4Instituto Keizo Asami, Universidade Federal de Pernambuco, Recife 50740-600, Brazil; 5Programa de Pós-Graduação em Biologia Aplicada à Saúde, Centro de Biociências, Universidade Federal de Pernambuco, Recife 50740-600, Brazil; 6Departamento de Educação Física, Universidade Federal de Pernambuco, Recife 50740-600, Brazil; 7Department of Immunology, Institute of Biomedical Sciences, University of São Paulo, São Paulo 05508-000, Brazil; 8Núcleo de Ciências da Vida-NCV, Centro Acadêmico do Agreste—CAA, Universidade Federal de Pernambuco, Caruaru 55014-900, Brazil

**Keywords:** inflammation, enriched environment, cytokines, macrophages, blood, brain, bone marrow

## Abstract

Environmental Enrichment (EE) is based on the promotion of socio-environmental stimuli, which mimic favorable environmental conditions for the practice of physical activity and health. The objective of the present systematic review was to evaluate the influence of EE on pro-and anti-inflammatory immune parameters, but also in cell activation related to the innate and acquired immune responses in the brain and peripheral tissues in rodents. Three databases [PubMed (2209 articles), Scopus (1154 articles), and Science Direct (1040 articles)] were researched. After applying the eligibility criteria, articles were selected for peer review, independently, as they were identified by September 2021. The protocol for this systematic review was registered in the PROSPERO. Of the 4417 articles found, 16 were selected for this systematic review. In the brain, EE promoted a reduction in proinflammatory cytokines and chemokines. In the blood, EE promoted a higher percentage of leukocytes, an increase in CD19+ B lymphocytes, and the proliferation of Natura Killer (NK cells). In the bone marrow, there was an increase in the number of CD27− and CD11b+ mature NK cells and a reduction in CD27− and CD11b+ immature Natural Killer cells. In conclusion, EE can be an immune modulation approach and plays a key role in the prevention of numerous chronic diseases, including cancer, that have a pro-inflammatory response and immunosuppressive condition as part of their pathophysiology.

## 1. Introduction

The Environmental Enrichment (EE) paradigm was first postulated by Hebb in 1947 to elucidate natural behavioral changes in laboratory animals, associated with possible changes in neurodevelopment [1]. This model is based on the promotion of inanimate and socio-environmental stimuli, which mimics favorable environmental conditions for the practice of physical activity in humans [2,3]. In this sense, it is known that unfavorable environmental conditions can promote physiological and psychological stress that is related to negative systemic changes, including those observed in the immune system [4,5,6].

The immune system fights microorganisms and toxins responsible for producing stressful, inflammatory responses and host infections. For this, the body uses two types of immune responses, (1) innate and (2) acquired. The innate response acts through several mechanisms, such as blocking the entrance of microorganisms in the epithelial surfaces, but also the proliferation, maturation, and targeting of neutrophils, basophils, eosinophils, mast cells, natural killers, and dendritic cells. Moreover, it is known the innate response engineers most of the pro-inflammatory cytokines, including the tumor necrosis factor-alpha (TNF-α), Interleukin-1 (IL-1), Interleukin-7 (IL-7), Interleukin-12 (IL-12), Interleukin-15 (IL-15), and Type 1 Interferon (IFN) and cytotoxicity [7,8]. On one hand, acquired immunity acts to contain these damages through B and T lymphocyte activation, capable of producing active responses, respectively [9].

Recent studies have demonstrated the efficiency of different EE protocols in positive immunomodulation. Takai et al. (2019) used EE for 6 weeks in mice transplanted with tumor cells. After this period, greater cytotoxic activity of Natural Killer (NK) cells, responsible for an antitumor response, was observed [10]. Furthermore, benefits related to the reduction in several manifestations of stress have been found after intervention with EE, demonstrating that such a non-pharmacological tool is promising in the treatment of adverse clinical and mental conditions [11,12]. Despite that, the real effects promoted by different EE protocols on the immune system are still not completely clear. Therefore, this systematic review aims to evaluate the performance of EE on (1) pro- and anti-inflammatory parameters and (2) immune cell activity related to the innate and acquired immune response in different tissues in rodents.

## 2. Methods

This present study was performed following the Preferred Reporting Items for Systematic Review and Meta-Analysis (PRISMA). The protocol for this systematic review was registered in the International Prospective Register of SR-PROSPERO (registration number CRD42021266055).

### 2.1. Search Strategy

We conducted a systematic search in the PubMed, Scopus, and Science Direct databases of studies published until December/2021. The selected studies describe the possible effects of Environmental Enrichment on the immune system in different species of rodents, in addition, the search terms were considered appropriate based on the Medical Subject Headings database (MeSH terms). The search strategy is fully described by the database in Table 1.

### 2.2. Selection of Articles

Articles were selected in two stages. In the first stage, two authors (GCJS and MSSF) independently assessed the titles and abstracts of each article found. Then, for abstracts that contained information according to the inclusion and exclusion criteria, the full text was read to observe the presence or absence of the eligibility criteria. Duplicates were automatically removed by creating an EndNote library. Possible discrepancies between evaluators were resolved by consensus (Figure 1).

Studies that met the eligibility criteria for the PICOS criteria were included in the study (Table 2): (1) **P**opulation: Rodents; (2) **I**ntervention: Environmental Enrichment; (3) **C**omparisons: Rodents exposed and not exposed to Environmental Enrichment; (4) **O**utcomes: Immune system activity: (I) Cytokine’s production (pro- and anti-inflammatory), (II) Natural killer, (III) Neutrophils, (IV) Macrophages, (V) Monocytes, (VI) T and B lymphocytes, among others. (5) Study design: rodent studies. Articles were excluded if (1) they do not present an environmental enrichment group, (2) the inclusion of samples with associated diseases or pharmacological or nutritional intervention, (3) reviews, opinions, letters, and human studies, or (4) unavailable text.

### 2.3. Quality Assessment

The SYRCLE’s strategy was used to assess the methodological quality of the animal studies. The tool consisted of ten questions that evaluate methodological criteria: Q1—Was the allocation sequence adequately generated and applied? Q2—Were the groups similar at baseline or were they adjusted for confounders in the analysis? Q3—Was the allocation to the different groups adequately concealed? Q4—Were the animals randomly housed during the experiment? Q5—Were the caregivers and/or investigators blinded from knowledge of which intervention each animal received during the experiment? Q6—Were the animals selected at random for the outcome assessment? Q7—Was the outcome assessor-blinded? Q8—Were incomplete outcome data adequately addressed? Q9—Are reports of the study free of selective outcome reporting? Q10—Was the study free of other problems that could result in a high risk of bias? Questions were answered with options of ‘Yes’, ‘No’, or ‘Not clear’. When the answer was ‘yes’, a score was given; when the answer was ‘no’ or ‘not clear’, no score was given. The overall scores for each article were calculated as a score of 0–10 points, with the quality of each study being classified as high (8–10), moderate (5–7), or low (<5). The two reviewers independently reviewed all included studies. Discrepancies between the evaluators were resolved by consensus. The quality outcomes are described in Table 3.

## 3. Results and Discussion

### 3.1. Characterization of Included Studies

In total, 4417 studies were selected from the databases: Pubmed/Medline (2209), Scopus (1154), and Science Direct (1054). However, 2495 were removed as they were duplicates and 1872 were excluded as they did not meet the eligibility criteria. The removal of duplicates was performed with the help of the EndNote^®^ software. Finally, 16 studies were included in the review (Figure 1).

Of the studies included, most studies were published between 2016 and 2020 (13 studies), using the mice strain C57BL/6N (six studies) with animals of both sexes and ages ranging from 3 weeks to 18 months. The experimental protocol divided the animals into groups of 4 to 24 animals per cage. The cage dimensions were reported in 12 studies. To compose the EE, different objectives were used, such as running wheels, tunnels, and toys. The EE protocols had a duration ranging from 1 week to 10 months. The description of the study protocols is found in (Table 4).

### 3.2. Cytokine Outcomes from Cells or Tissues after EE Intervention

#### 3.2.1. Pro/Anti-Inflammatory Cytokines Levels and Their Receptors in Different Brain Regions

The production of pro and anti-inflammatory cytokines was observed in 12 studies. In the brain, cytokines and chemokines were produced in different regions such as the hypothalamus, amygdala, hippocampus, and Pre-Frontal Cortex (PFC). Ali et al. (2019) observed that the hypothalamus significantly reduced CCL2, IL-1β, and IL-6 gene expression [13]. Similarly, a decrease in CCL2 and NFkB was observed in the amygdala. However, there were no significant differences in the levels of IL-1β and IL-6 in the groups that were exposed to EE [13]. In the hippocampus, a significant decrease in CCL2, IL-1β, IL-6, CXCL10, IL-10, IFN-γ, TNF-α, and MCP-1 was observed [16,21,23,24,25]. In those five studies, EE was not able to modify the levels of IL-6, IL-10, IL-1β, and TNF-α in the hippocampus [16,18,20,23,26], (Table 4).

In addition to the EE, it was observed that the animals were exposed to stressful factors, such as social isolation and swimming tests. In PFC, Wang et al. (2018) and McQuaid et al. (2013) have noticed a reduction in IL-1β and TNF-α, but no differences were observed in IL-6 [20,23]. Additionally, McQuaid et al. (2013) have remarked that a significant decrease in IL-6 receptor (IL-6r), IL-1 receptor (IL-1r), and TNFr1b were observed in the hippocampus [20]. In PFC, no differences were observed in the levels of IL-6r, IL-1r, TNF-α receptor (TNF-αr), and proIL-1β [20] (Figure 2).

#### 3.2.2. Pro/Anti-Inflammatory Cytokines Levels in the Peritoneal Cavity, Spleen, and Lymphocytes

Levels of pro-inflammatory cytokines were also observed in the Peritoneal Cavity (PC), spleen, and lymphocytes. Brod et al. (2017) have observed non-significant differences in the PC in TNF-α and MCP-1 levels after 2 weeks of EE protocol [15]. In another protocol lasting 6–8 weeks, a significant increase in MCP-1 and IL-6 expression was observed [27]. Moreover, Arranz et al. (2010) investigated rodents exposed to 6–8 weeks of EE. It was also possible to identify increased levels of IL-2 and TNF-α in the PC [14]. In the spleen, no differences were found in the levels of IL-2, IL-4, IL-10, and IFN-γ. However, when the ratios between IFN-γ/IL-10 and IL-2/IL-10 were analyzed, Marashi et al. (2003) noted a significant reduction in their levels after EE [19]. Regarding cytokine production by immune cells, Rattazi et al. (2016) did not observe significant differences in the IL-4, IL-10, and IL-17 levels, but there was a significant reduction in IFN-γ production by the lymphocytes [22], (Table 5).

### 3.3. Immune Cell Compartment after Intervention with EE in Different Tissues

The presence of cells that compose the immune system was observed in 10 studies. The impact of EE was investigated through immune cells in different tissues such as the hippocampus, PC, spleen, lymph nodes, bone marrow, and blood.

#### 3.3.1. Hippocampus and Peritoneal Cavity

In the hippocampus, Zhang et al. (2017) noticed a significant decrease in CD68+ cells classified as macrophages in mice that were exposed to EE [26]. In the PC, there was a significant decrease in the percentage of CD4 T cells and a significant increase in the percentage of CD11b+ cells (leukocyte differentiation antigen typical of macrophages) and NK cells [14]. In addition, high levels of expression of Toll-like receptor (TLR) 2 and TLR4 in CD11b+, CD11c+ (dendritic cells), CD4 (T helper lymphocytes), CD8 (T cytotoxic lymphocytes), and CD19+ cells (B lymphocytes) were found. Nonetheless, there was a significant decrease in the percentage of CD4 and CD11c+ cells expressing TLR2 and TLR4. In CD8 T cells, a similar reduction in TLR2 was found after EE [14]. Besides, a higher efficiency rate was observed in macrophage phagocytosis and a significant increase in lymphocyte chemotaxis in animals submitted to EE was also identified [14,27] (Table 5).

#### 3.3.2. Blood

In the blood, EE promoted a higher percentage of leukocytes, a significant increase in CD19+ B lymphocytes, and the proliferation of NK cells [15,17,21]. However, Gurfein et al. (2017) observed no significant differences for CD4, CD11b+, CD49b+, NK, and Ly6G+ cells [17]. A decrease in the ratio between B/T lymphocytes and a reduction in CD8 T cells in protocols with social interaction was also observed [19]. In the blood, there was a significant increase in the number of CD27-CD11b+ mature NK cells and a reduction in CD27+ and CD11b+ immature NK cells.

#### 3.3.3. Bone Marrow and Lymphoid Tissues

In the bone marrow, no significant difference in the percentage of NK cells was found [21,28]. Nonetheless, Meng et al. (2018) remarked a significant increase in the percentage of NK cells in the spleen. However, also, there was an increase in proliferation in both tissues [21]. Immature CD27+ and CD11b+ mature NK cells were investigated only in the study by Meng et al. (2018), which evaluated the presence in bone marrow and spleen. In all tissues, there was an increase in the number of CD27-CD11b+ mature NK cells and a reduction in CD27− and CD11b+ immature NK cells [21,28]. In the spleen, significant differences were not observed in the number of CD8, CD3, and CD4 T cells [22,25]. However, Rattazi et al. (2016) found a decreased differentiation of CD4 T cells in the spleen, thymus, and lymph nodes [22].

### 3.4. Methodological Quality of Studies

When evaluating the individual items of the SYRCLE tool, it was observed that no study made it clear how the allocation sequence was performed. Nevertheless, all studies used groups at the beginning of the experimental protocol, as well as the allocation of animals and housing, which were equally implemented. Due to the EE protocol structure, the animals in each group were randomly selected, although, none of the studies performed outcome selection and did not have a high risk of bias. All the articles included had a score of 6, thus being classified as moderate quality and suitable for composing the present systematic review. The results are described in Table 3.

This systematic review aimed to summarize the findings in the literature about the effect of EE on pro- and anti-inflammatory parameters and immune cell activity related to the innate and acquired immune response in different tissues in rodents. It was verified that the exposure to different EE protocols (the number of rodents, the length, width, depth of the cage, and the time of exposure) was able to decrease the levels of inflammatory markers, followed by a possible anti-inflammatory response. Additionally, we found that cellular responses may vary according to the tissue, morphology, and functionality; among them are the increase in macrophage phagocytosis, NK cell, and lymphocyte activity.

It is widely known that the increase in inflammatory processes is related to the establishment of cognitive and behavioral dysfunctions and the emergence and clinical progression of neurodegenerative diseases such as depression, Parkinson’s, and Alzheimer’s Diseases [29,30,31]. In this sense, several studies have demonstrated the impact of EE on the immune response in different brain areas, mainly in the hippocampus, which effectively participates in emotional control and memory capacity. Moreover, this reduction in inflammatory markers in the hippocampus, including CCL-2, CXCL10, IL-1β, IL-6, IFN-γ, and TNF-α, is essential to prevent the reduction in neuron synthesis, proliferation, and growth in different regions such as dentate Gyrus and *Cornu Ammonis* 1(CA1), 2 (CA2), and 3 (CA3) in rodents [32,33,34,35]. Furthermore, it is important to highlight that those inflammatory responses depend on signal transduction through receptors. Different regions expressed different levels of receptors (e.g., IL-6R, IL-1R, and TNF-αR). This demonstrates that each region may have specific responses in immune responses and outcomes.

Next, we could observe inflammatory response modulation by EE in peripheral tissues (PC and spleen) and lymphocytes. The PC has a visceral lining that is highly sensitive to various internal and external stimuli influencing the immune system activity [36]. Our analysis demonstrates a discrepancy in the expression of pro-inflammatory cytokines in mice PC after an EE protocol. In EE protocols from 6 to 8 weeks, an increase in the expression of IL-2, TNF-α, IL-6, and MCP-1 in PC was observed.

Studies have shown that higher levels of these inflammatory markers, mainly IL-6 and TNF-α, are related to peritonitis and other abdominal infections [37,38]. However, the authors argue that these biological responses may have been caused by the difference in the duration of exposure to EE, as minimal or acute exposure (2 weeks) to EE was able to promote positive immunomodulatory effects and antitumor response in other studies [10]. The spleen is a fundamental organ for blood filtration and the removal of damaged erythrocytes, and it actively participates in the immune response. When analyzed in different studies, it was possible to find decreased relationships between IFN-γ/IL-10 and IL-2/IL-10.

In immune cells, different tissue responses were also observed after EE. For example, in the hippocampus, Zhang et al. (2017) [26] observed a reduction in CD68+ cells after 3 weeks of EE. The CD68 protein is highly expressed by macrophages which are exploited as a cytochemical marker of monocytes in immunostaining assays and associated with other macrophage-related biomarkers playing a fundamental role in cancer histopathology [39]. In addition, studies have shown that increased CD68+ cell activation is involved in microglial neuroinflammation and hyperactivation, which also occurs due to increased infiltration of M1 macrophages, directly affecting the hippocampus through dysfunctions in memory capacity, which is part of the etiology of several neurodegenerative diseases [40,41,42,43].

In PC, different outcomes in the expression of cells and TLRs were found after EE. Arranz et al. (2010) [14] observed a reduction in the percentage of CD4 T cells in all groups after EE in mice with different age patterns (adults, elderly, and very old). Low levels of CD4 T cells are related to a greater vulnerability to immunosuppression and are present in several infections, including the human immunodeficiency virus (HIV) [44]. The changes in CD4 may have been caused by the natural aging process, which can generally increase the susceptibility to infections [45]. On the other hand, in very old mice (VO), high percentages of CD11b+ and NK cells were found after EE. Evidence points out that high levels of these cells promote greater effectiveness by cytotoxic combating pathogens, demonstrating that EE positively modulates the immune system containing the deleterious effects of aging.

After 10 months of EE, age-independent variability in the expression of TLR2 and TLR4 in the lymphocyte cell membrane was observed. These receptors play a fundamental role in the activation of innate immunity, recognizing specific microbial patterns. In addition, they can be modulated by aging and are associated with the production of pro-inflammatory cytokines, cardiometabolic diseases, and cancer [46,47]. In CD8 lymphocytes, there was a reduction in TLR2 expression. Furthermore, in CD4 and CD11c +, lower levels of TLR2 and TLR4 were also observed after EE in VO mice. These cells are important in structuring the immune response against several pathogens, especially in the antiviral and tumor response. Patients with acquired immunodeficiency syndrome (AIDS) present immunosuppression associated with low levels of CD4/CD8, providing a worse clinical prognosis and higher mortality rate. Furthermore, a significant reduction in TLR2 and TLR4 expression in these cells is linked to anti-inflammatory cytokine activity and better health conditions [48].

Then, there was a contrast in the immunological responses promoted by EE in the blood. A higher percentage of leukocytes, a proliferation of NK cells, and an increase in CD11b+ and CD27 mature lymphocytes were observed. These findings demonstrate that the cells that composers compose the immune system are positively modulated by EE. When there is a lack of effectiveness in the actions of these components, the body becomes more susceptible to allergies, inflammation, infections, and neoplasms [49,50].

On the other hand, using EE and social interaction, lower rates of B/T and CD8 lymphocytes were also found, and the authors discuss behavioral aspects including the hierarchy between male mice and aggressiveness and increased corticosterone and adrenal tyrosine hydroxylase, which may explain the decrease in CD8 after EE. A limitation of the study by Marashi et al. (2003) [19] is related to the time in which the EE was applied, which was not described in the study; thus, it may have influenced the synthesis of the results.

Finally, cellular changes in lymphoid organs were observed after EE. The NK cell activity and proliferation increases after two weeks of EE were identified. These cells in the bone marrow have different roles, including chemotaxis and homeostatic proliferation during viral infections. Recent studies have shown the participation of NK cells against infections and tumors [51]. Furthermore, the increases in CD11b+ leukocytes and CD27+ lymphocytes emerge as a potential therapeutic target in the synthesis of anticancer drugs [50], although, in both the thymus and lymph nodes, there was a reduction in the differentiation of CD4 lymphocytes after two weeks of EE. In these organic components occur massive lymphocyte traffic and maturation, essential for the stability of the immune system in cases of aggression and to maintain homeostasis through multiple factors. Therefore, all these results point out the ability of the EE to act on the immune system and its components, but a causal investigation of the EE in these markers is necessary to establish an effective relationship.

The limitations of this systematic review mainly include EE structures (durations in weeks and months, number of animals, number, type of toys, and other elements in the cage), which can promote variability in the effects found on the immune system. Regardless, we believe our work is significant as it systematically summarizes the available evidence for future research to consider. Furthermore, we recognize that there is heterogeneity in the quality of reference sources; therefore, higher-quality studies and rigorous methodology are needed. However, within the scientific literature eligible to synthesize this systematic review, we were able to observe the modulation of EE in different types of outcomes related to the immune response through several markers that helped us to confirm our research hypothesis that EE would be able to promote positive modulation in the innate and acquired immune response.

## 4. Conclusions

In conclusion, this is the first systematic review that identified the positive impacts of EE in the pro- and anti-inflammatory aspects and cell activity related to the innate and acquired immune responses in different tissues in rodents. Additionally, this study highlights the EE as an emerging non-pharmacological tool to combat dysregulations in the homeostasis of the immune system, including hyperinflammatory response and immunosuppression.

## Figures and Tables

**Figure 1 ijms-23-11986-f001:**
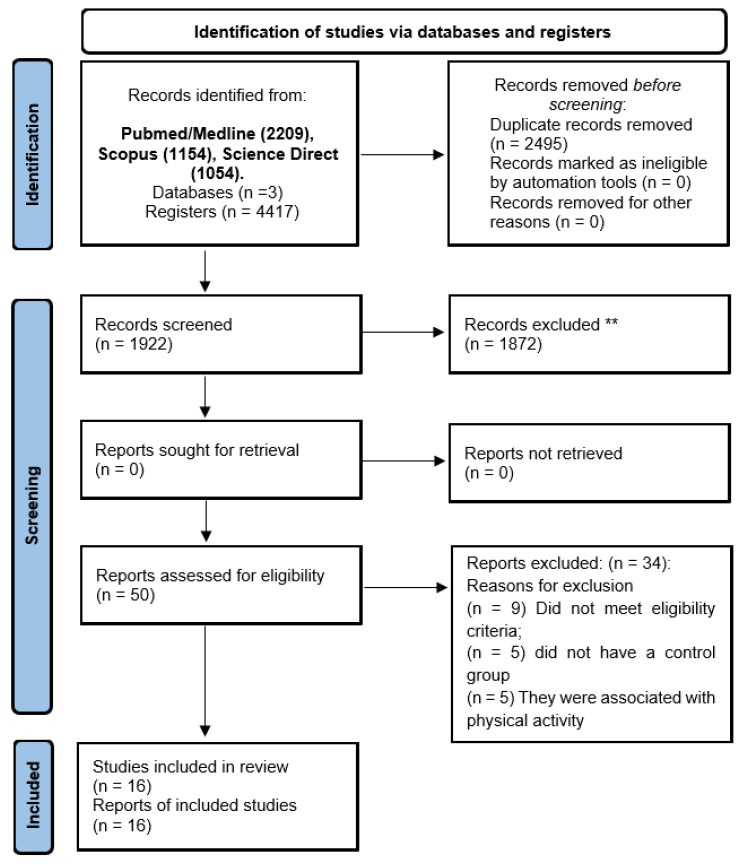
PRISMA 2020 flow diagram for new systematic reviews which included searches of databases and registers only. Consider, if feasible to do so, reporting the number of records identified from each database or register searched (rather than the total number across all databases/registers); ** If automation tools were used, indicate how many records were excluded by a human and how many were excluded by automation tools.

**Figure 2 ijms-23-11986-f002:**
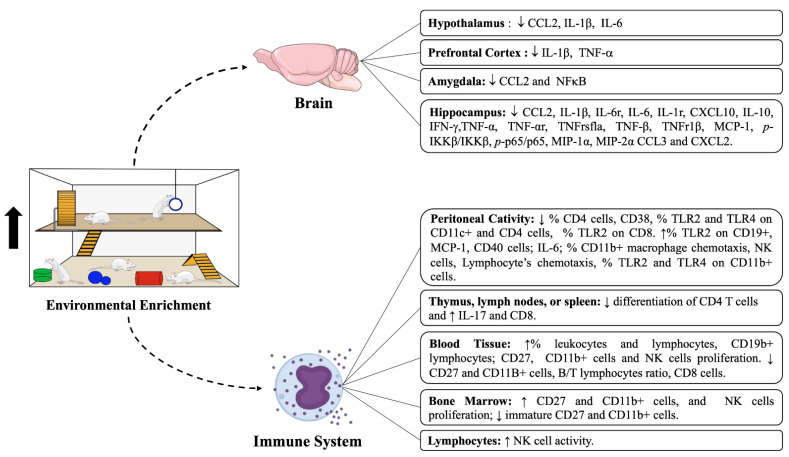
Impacts of environmental enrichment protocols on the activity of different cells of the immune system and expression of pro and anti-inflammatory cytokines in central (Hypothalamus, Prefrontal Cortex, Amygdala, and Hippocampus) and peripheral tissues (Peritoneal cavity, Thymus, Lymph nodes, Spleen, Blood Tissue, or Bone Marrow) and Lymphocytes.

**Table 1 ijms-23-11986-t001:** Sample search strategy on the database.

Database	Code Line
Pubmed/Medline Scopus	(((Environmental Enrichment) OR (Enriched Environment)) AND ((((((((Immune System) OR (Immune Systems)) OR (System, Immune)) OR (Inflammation)) OR (Inflammations)) OR (Innate Inflammatory Response)) OR (Cytokine)) OR (Cytokines)).
ScienceDirect	(((“Exercise”) OR (“Physical Activity”)) OR (“Physical Exercise”)) AND ((((“Endoplasmic Reticulum Stress”) OR (“Endoplasmic Reticulum Stresses”)) OR (“Stress, Endoplasmic Reticulum”))

**Table 2 ijms-23-11986-t002:** Eligibility criteria based on the PICOS strategy.

	Inclusion Criteria	Exclusion Criteria
**P**opulation	Rodents	Humans and other species
**I**ntervention	Environmental Enrichment	Absence of the Environmental Enrichment
**C**omparator	Animals that did not undergo Environmental Enrichment	Absence of the control group
**O**utcomes	Cytokines and immune cells	No Cytokines and immune cells
**S**tudy design	Experimental	Observational, reviews, case reports, and scientific abstracts

**Table 3 ijms-23-11986-t003:** Methodological Quality of Studies using the SYRCLE strategy.

Study	Q1	Q2	Q3	Q4	Q5	Q6	Q7	Q8	Q9	Q10	Score
Ali et al., 2019 [13]	U	Y	Y	Y	N	Y	U	U	Y	Y	6
Arranz et al., 2010 [14]	U	Y	Y	Y	N	Y	U	U	Y	Y	6
Brod et al., 2017 [15]	U	Y	Y	Y	N	Y	U	U	Y	Y	6
Ferré et al., 2016 [16]	U	Y	Y	Y	N	Y	U	U	Y	Y	6
Gurfein et al., 2017 [17]	U	Y	Y	Y	N	Y	U	U	Y	Y	6
Keymoradzadeh et al., 2020 [18]	U	Y	Y	Y	N	Y	U	U	Y	Y	6
Marashi et al., 2003 [19]	U	Y	Y	Y	N	Y	U	U	Y	Y	6
McQuaid et al., 2013 [20]	U	Y	Y	Y	N	Y	U	U	Y	Y	6
Meng et al., 2018 [21]	U	Y	Y	Y	N	Y	U	U	Y	Y	6
Rattazi et al., 2016 [22]	U	Y	Y	Y	N	Y	U	U	Y	Y	6
Takai et al., 2019 [10]	U	Y	Y	Y	N	Y	U	U	Y	Y	6
Wang et al., 2018 [23]	U	Y	Y	Y	N	Y	U	U	Y	Y	6
Williamson et al., 2012 [24]	U	Y	Y	Y	N	Y	U	U	Y	Y	6
Zarif et al., 2017 [25]	U	Y	Y	Y	N	Y	U	U	Y	Y	6
Zhang et al., 2017 [26]	U	Y	Y	Y	N	Y	U	U	Y	Y	6

Notes: Q1: Was the allocation sequence adequately generated and applied? Q2: Were the groups similar at baseline or were they adjusted for confounders in the analysis? Q3: Was the allocation to the different groups adequately concealed during? Q4: Were the animals randomly housed during the experiment? Q5: Were the caregivers and/or investigators blinded from knowledge of which intervention each animal received during the experiment? Q6: Were the animals selected at random for the outcome assessment? Q7: Was the outcome assessor blinded? Q8: Were incomplete outcome data adequately addressed? Q9: Are reports of the study free of selective outcome reporting? Q10: Was the study apparently free of other problems that could result in high risk of bias? Unclear; Y: Yes; N: No.

**Table 4 ijms-23-11986-t004:** Sample description and characterization of environmental enrichment protocols and Brain outcomes.

Author, Year	Species/Sex and Age	Environmental Enrichment Protocol, Animals Data, and Housing Dimensions (Length, Width, and Depth)	Exposure Time to Environmental Enrichment	Brain Outcomes
Ali, 2019 [13]	C57BL/6N mice, Female, and 10 months old	Wheels, tunnels, igloos, huts, retreats, wood toys, and a maze in addition to standard chow and water. n = 5 and 63 cm × 49 cm × 44 cm	10 months	**Hypothalamus:** ↓ CCL2, IL-1*β*, IL-6. **Amygdala:** ↓ CCL2 e NFkbia/= IL-1*β*, IL-6, SOCS3.
Arranz, 2010 [14]	ICR/CD-1 mice, Female, Adults: 44 ± 4 weeks, Old: 69 ± 4 weeks, Very Old: 92 ± 4 weeks	Range bucket, jolly ball, hoop, holed ball, yellow tunnel, rough red object, yellow billiard ball, and silver ball. Uninformed and Uniformed.	6–8 weeks	-
Brod, 2017 [15]	CD1 mice, male and 6 weeks old	Wheel, nest house, tunnel, ample nesting material, and wood chip bedding to a 5 cm depth. Uninformed and 50 × 38 cm × 21 cm.	2 weeks	-
Ferré, 2016 [16]	SAMP8 mice, Male and 3 months old	Plastic tubes (20 cm long and 2.5 cm in diameter) were placed in EE cardboard cages, moreover to plastic dolls or toys, which were added, removed, or changed every week. Uninformed and Uninformed.	Uninformed	**Hippocampus:** ↓ IL-6; CXCL10/= TNF-α
Gurfein, 2017 [17]	BALB/c mice, Male and 6 weeks old	Uninformed, n = 10 and 257 mm × 483 mm × 152 mm = 980 cm^2^	9 weeks	-
Keymoradzadeh, 2020 [18]	Wistar Rats, Male and 7 weeks old	Running wheels, a tunnel, a small compartment, stairs, and many other colorful objects (e.g., colorful plastic plates, wooden disks of varied colors and sizes, plastic cups, and hanging cubes). n = 7 and 96 cm × 49 cm × 38 cm.	3 weeks	**Hippocampus:** ↓ IL-6; CXCL10/= TNF-α.
Marashi, 2003 [19]	Strain CS of the inbred strain ABG mice, Male and uniformed	Passable enriched cage, extra plains, plastic stairs, wooden footpaths, hemp ropes, and a climbing tree. n = 4 and 100 cm × 40 cm × 34.5 cm	Uninformed	**Hippocampus:** = IL-1*β* and IL-10.
Mcquaid, 2013 [20]	BALB/cByJ mice, male and 6–8 weeks old	Two running wheels, one red polypropylene shelter, one orange polypropylene shelter with an angled running wheel, as well as three yellow polypropylene tunnels and two cotton nestlets. Uninformed and 50 cm × 40 cm × 20 cm.	4 weeks	**Hippocampus:** = IL-6, IL-1b, TNF-α. ↓ IL-6r; IL-1r/= TNF-αr. **Prefrontal Córtex:** = IL-6r; IL-1r and TNF-αr.
Meng, 2018 [21]	C57BL/6N mice/Male/3 or 10 weeks old	Running wheels, wood toys, plastic tunnels, ladders, huts, and nesting materials. Toys were changed every 3–4 days. n = 12 and 61 cm × 43 cm × 21 cm.	5 weeks	-
Otaki, 2018 [27]	B6C3F1/Cr1 mice/Male and 3 weeks	Running wheels, tunnel, bio-hut, wood gnawing block, shelter, and nesting sheet. n = 5 and Uniformed.	6–8 weeks	-
Rattazi, 2016 [22]	C57BL/6N mice/Male and 6 weeks	Colored transparent plastic nest-box, fabric tube, running wheel, and one wood hamster swing. n = 6 and 40 cm × 25 cm × 20 cm.	2 weeks	-
Takai, 2019 [10]	B6C3F1/mice/Female and 6 weeks old	Uninformed, n = 8–24 and Uninformed.	6 weeks	-
Wang, 2018 [23]	C57BL/6N mice/Male and 18 months old	Uninformed, n = 4 and 47 cm × 30 cm × 23 cm	2 months	**Hippocampus:** ↓ IL-1b; TNF-α; =IL-6; =pro-IL1b; ↓ *p*-IKKβ/IKKβ and *p*-P65/P65. **Prefrontal Cortex:** ↓ IL-1b; TNF-α; = IL-6; = pro-IL1β/= *p*-IKKβ/IKKβ and *p*-P65/P65.
Williamson, 2012 [24]	Sprague -Dawley	Quarter-inch corn-cob bedding (identical to home cage controls), a running wheel, a PVC tube, and various small objects and toys. n = 6–8; 55.9 cm × 35.6 cm × 30.5 cm	7 weeks	**Hippocampus:** ↓ IL-1β; TNF-α; TNF-rsfla; TNF-r1β; TNF-β; ↓ MCP-1/CCL2; MIP-1α/CCL3; MIP-2α/CXCL2.
Zarif, 2017 [25]	C57BL/6N mice/Female and 4 weeks	Nesting material, houses, running wheels, hammocks, scales, plastic toys, and tunnels. n = 5–6, 120 cm × 76 cm × 21 cm	1–4 weeks	**Hippocampus:** ↓ TNF-α; = IFN-γ.
Zhang, 2017 [26]	C57BL/6 N mice and 10 weeks old	Plastic tunnels, wooden climbing frame, platforms, hiding shelters, house, exercise wheel, chew toys, and other novel objects designed specifically for small animals = 20 and 56 cm × 40 cm × 22 cm.	3 weeks	**Hippocampus:** = IL-1*β*; TNF-α; ↓ CD68+

Notes: cm: centimeters, CCL2: Chemokine C-C Ligand 2; CCL3: Chemokine C-C Ligand 3; CXCL2: C-X-C Chemokine Ligand 2; CXCL10: C-X-C Chemokine Ligand 10; IKKβ: Inhibitor of Nuclear Factor Kappa-B Kinase Subunit Beta; *p*-IKKβ: Inhibitor of Nuclear Factor Kappa-B Kinase Subunit Beta Phosphorylated; IL-1b: Interleukin-1 Beta, pro-IL-1b: pro-Interleukin-1 Beta; IL-1r: Interleukin-1 receptor; IL-6: Interleukin-6; IL-6r: Interleukin-6 receptor; IL-10: Interleukin-10; Lymphocyte antigen 6 complex, locus g; MCP-1: Monocyte Chemoattractant Protein-1; NFkbia: Nuclear Factor Kappa B inhibitor alpha; P65: Protein 65; p-P65: Protein 65 Phosphorylated; SOCS3: Suppressor of Cytokine Signaling 3; TNF-α: Tumor Necrosis Factor-alpha; TNF-β: Tumor Necrosis Factor-Beta;TNF-r1β: Tumor Necrosis Factor receptor 1 Beta; TNF-rsfla: Tumor Necrosis Factor receptor superfamily member 1A. = There were no significant differences.

**Table 5 ijms-23-11986-t005:** Sample description and characterization of environmental enrichment protocols and Cytokines and Immune Cells in Peripheral Tissues.

Author, Year	Species/Sex and Age	Environmental Enrichment Protocol, Animals Data, and Housing Dimensions (Length, Width, and Depth)	Exposure Time to Environmental Enrichment	Peripheral Outcomes
Ali, 2019 [13]	C57BL/6N mice, Female, and 10 months old	Wheels, tunnels, igloos, huts, retreats, wood toys, and a maze in addition to standard chow and water. n = 5 and 63 cm × 49 cm × 44 cm	10 months	**Peritoneal Cavity:** ↓ % CD4 cells in all groups. ↑ % CD11b+ in very old mice (VO). ↑ macrophage chemotaxis in VO. ↑ phagocytic index in old and VO; ↑ phagocytic efficacy in VO. ↑ NK cells proliferation in adult and VO. ↑ NK cells activity in VO. ↑ Lymphocyte’s chemotaxis in old and VO. ↑ % TLR2 and TLR4 on CD11b+ cells in VO. ↓ % TLR2 and TLR4 on CD11c+ and CD4 cells in VO. ↓ % TLR2 on CD8 cells in VO. ↑% TLR2 on CD19+ in VO.
Arranz, 2010 [14]	ICR/CD-1 mice, Female, Adults: 44 ± 4 weeks, Old: 69 ± 4 weeks, Very Old: 92 ± 4 weeks	Range bucket, jolly ball, hoop, holed ball, yellow tunnel, rough red object, yellow billiard ball, and silver ball. Uninformed and Uniformed.	6–8 weeks	**Blood Tissue:** ↑ % leukocytes and lymphocytes. Peritoneal Cavity: ↑ IL-2/↑ TNF-α, = IL-6, MCP-1 and TNF-α.
Brod, 2017 [15]	CD1 mice, Male and 6 weeks old	Wheel, nest house, tunnel, ample nesting material, and wood chip bedding to a 5 cm depth. Uninformed and 50 cm × 38 cm × 21 cm.	2 weeks	-
Ferré, 2016 [16]	SAMP8 mice, Male and 3 months old	Plastic tubes (20 cm long and 2.5 cm in diameter) were placed in EE cardboard cages, and plastic dolls or toys, which were added, removed, or changed every week. Uninformed and Uninformed.	Uninformed	**Blood Tissue:** ↑CD19b+ lymphocytes; = Thy1.2+T lymphocytes; = CD4+T lymphocytes; = CD8+ lymphocytes; = CD11b+ monocytes; CD49b+ NK cells; Ly-6g+ neutrophils; ↓ B/T lymphocytes ratio.
Gurfein, 2017 [17]	BALB/c mice, Male and 6 weeks old	Uninformed, n = 10 and 257 mm × 483 mm × 152 mm = 980 cm^2^	9 weeks	-
Keymoradzadeh, 2020 [18]	Wistar Rats, Male and 7 weeks old	Running wheels, a tunnel, a small compartment, stairs, and many other colorful objects (e.g., colorful plastic plates, wooden disks of varied colors and sizes, plastic cups, and hanging cubes). n = 7 and 96 cm × 49 cm × 38 cm.	3 weeks	**Blood Tissue:** = CD4/CD8 ratio and CD4; ↓ CD8 cells. **Spleen:** = IL-2; IL-4; IL-10 and IFN-γ/↓ IFN-γ/IL-10; ↓IL-2/IL-10.
Marashi, 2003 [19]	Strain CS of the inbred strain ABG mice, Male and uniformed	Passable enriched cage, extra plains, plastic stairs, wooden footpaths, hemp ropes, and a climbing tree. n = 4 and 100 cm × 40 cm × 34.5 cm	Uninformed	-
Mcquaid, 2013 [20]	BALB/cByJ mice, male and 6–8 weeks old	Two running wheels, one red polypropylene shelter, one orange polypropylene shelter with an angled running wheel, as well as three yellow polypropylene tunnels and two cotton nestles. Uninformed and 50 cm × 40 cm × 20 cm.	4 weeks	**Bone marrow:** = NK cells; ↑ mature CD27 and CD11b+ cells; ↓ immature CD27 and CD11b+ cells. ↑ NK cells proliferation. **Spleen:** ↑ % NK cells; ↑ mature CD27 and CD11b+ cells; ↓immature CD27 and CD11b+ cells. Blood Tissue: = NK cells; ↑ CD27 and CD11b+ cells; ↓ immature CD27 and CD11B+ cells. ↑ NK cells proliferation.
Meng, 2018 [21]	C57BL/6N mice/Male/3 or 10 weeks old	Running wheels, wood toys, plastic tunnels, ladders, huts, and nesting materials. Toys were changed every 3–4 days. n = 12 and 61 cm × 43 cm × 21 cm.	5 weeks	**Peritoneal Cavity:** ↑ macrophage phagocytosis. = MIP-2; ↑ MCP-1; ↑ Phagocytosis of apoptotic neutrophils. M1 markers: ↑ CD40 cells; ↑ IL-6; =NOS2 and ↓ CD38.
Otaki, 2018 [27]	B6C3F1/Cr1 mice/Male and 3 weeks	Running wheels, tunnel, bio-hut, wood gnawing block, shelter, and nesting sheet. n = 5 and Uniformed.	6–8 weeks	**Thymus, lymph nodes, or spleen:** = CD3 and CD4 T cells; ↓ differentiation of CD4 T cells in response to IFN-γ and ↑ IL-17. Lymphocytes: = IL-4; IL-10; ↑ IL-17; ↓ IFN-γ
Rattazi, 2016 [22]	C57BL/6N mice/Male and 6 weeks	Colored transparent plastic nest-box, fabric tube, running wheel, and one wood hamster swing. n = 6 and 40 cm × 25 cm × 20 cm.	2 weeks	**Lymphocytes:** ↑ NK cell activity; = NK cells.
Takai, 2019 [10]	B6C3F1/mice/Female and 6 weeks old	Uninformed, n = 8–24 and Uninformed.	6 weeks	-
Wang, 2018 [21]	C57BL/6N mice/Male and 18 months old	Uninformed, n = 4 and 47 cm × 30 cm × 23 cm	2 months	-
Williamson, 2012 [24]	Sprague -Dawley	Quarter-inch corn-cob bedding (identical to home cage controls), a running wheel, a PVC tube, and various small objects and toys. n = 6–8; 55.9 cm × 35.6 cm × 30.5 cm	7 weeks	**Spleen:** = CD8+ T cells; = CD8/CD4; ↑ CD8 cell proliferation
Zarif, 2017 [25]	C57BL/6N mice/Female and 4 weeks	Nesting material, houses, running wheels, hammocks, scales, plastic toys, and tunnels. n = 5–6, 120 cm × 76 cm × 21 cm	1–4 weeks	**Peritoneal Cavity:** ↓ % CD4 cells in all groups. ↑ % CD11b+ in very old mice (VO). ↑ macrophage chemotaxis in VO. ↑ phagocytic index in old and VO; ↑ phagocytic efficacy in VO. ↑ NK cells proliferation in adult and VO. ↑ NK cells activity in VO. ↑ Lymphocyte’s chemotaxis in old and VO. ↑ % TLR2 and TLR4 on CD11b+ cells in VO. ↓ % TLR2 and TLR4 on CD11c+ and CD4 cells in VO. ↓ % TLR2 on CD8 cells in VO. ↑% TLR2 on CD19+ in VO.
Zhang, 2017 [26]	C57BL/6N mice/Male and 10 weeks old	Plastic tunnels, wooden climbing frame, platforms, hiding shelters, house, exercise wheel, chew toys, and other novel objects designed specifically for small animals = 20 and 56 cm × 40 cm × 22 cm.	3 weeks	**Blood Tissue:** ↑ % leukocytes and lymphocytes. **Peritoneal Cavity:** ↑ IL-2/↑ TNF-α, = IL-6, MCP-1 and TNF-α.

Notes: cm: centimeters, CD4: Cluster of Differentiation 4; CD8: Cluster of Differentiation 8; CD11b+: Cluster of Differentiation 11 b+; CD19b+: Cluster of Differentiation 19 b+; CD27: Cluster of Differentiation 27; CD38: Cluster of Differentiation 38; CD49b+: Cluster of Differentiation 49 b+; IL-2: Interleukin-2; IL-4: Interleukin-4; IL-6: Interleukin-6; IL-10: Interleukin-10; IL-17: Interleukin-17; IFN-γ: Interferon-gamma; Ly-6g+:Lymphocyte antigen 6 complex, locus g; MCP-1: Monocyte Chemoattractant Protein-1; NK: Natural Killer Cell; NOS2: Nitric Oxide Synthase 2; TLR2: Toll-Like Receptor-2; TLR4: Toll-Like Receptor-4; TNF-α: Tumor Necrosis Factor-alpha;. = There were no significant differences.

## Data Availability

All data from the articles used in this systematic review are available in the references.

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
