# Peer review of "Cytokines and Immune Cells Profile in Different Tissues of Rodents Induced by Environmental Enrichment: Systematic Review"

_ijms, 2022, doi:10.3390/ijms231911986_

Round 1
Reviewer 1 Report
Int. J of Mol Sci
Systematic Review. Cytokines and immune cells profile in different tissues of rodents induced by Environmental Enrichment
Matheus Santos de Sousa Fernandes et al.
Novel experiences influence stem/progenitor cell behavior in the adult rodent brain, and activity-dependent changes in neuroplasticity occur; including increased cell proliferation in hippocampus upon physical exercise and cell survival upon exposure to an enriched environment (ENR, EE). Living in ENR can also counterattack exposure to stress and reduce the release of pro-inflammatory factors. Depending on the microenvironment, cytokines function pro- (i.e., IL-1β, IL-6, IL-18, and TNF-α) or anti-inflammatory (i.e., IL-4, IL-10). The current systemic review summarize 16 peer reviewed articles on the effects of various times mice or rats have lived in ENR, and the impact on inflammation, both in brain and peripheral tissue, the release of pro/anti-inflammatory factors and macrophages, lymphocytes etc profile. The topic of the review is interesting, significant and comes out at the right time. It overall needs some structural corrections, and a more precisely mention of the reviewed papers, i.e. throughout the discussion; and editorial supervision. Please find my general comments on the order, and specific indications to improve the relevance for a broader audience below.
Abstract, Introduction
use ‘articles’ over ‘reports’; ‘influence, role, impact’ instead of ‘performance’, and ‘immune response’; distinguish ‘brain and peripheral tissue’ rather than ‘different tissue’; 1st citation is not Hebb but a review by Gerd Kempermann, it should be stated as that, Kempermann is also best known for the positive effects ENR has on neuroplasticity; in the Abstract, already write statements: “In the brain, EE ‘positively’ affects .. the modulation of cytokines and chemokines … or ‘decreases the release of pro-inflammatory factors’… ”. Please indicate what CD xx, NK cells etc. mean or use broader terms. For the Introduction, introduce the various pro, anti-inflammatory factors first (i.e. like above), briefly state the role of microglia, macrophages, lymphocytes etc., to make the overall impact of the findings of the 16 articles more relevant; why is only, or specifically, Takai et al. (2019) mentioned in the introduction, and not already a handful of the 16 papers. Why were the criteria for ‘science direct’ suddenly changed, please expend om that (page 4), and since ‘physical exercise’ was a disqualifier (see Fig. 1).
I suggest the following Keywords: inflammation (not immunity), enriched environment (ENR, EE), cytokines, macrophages, blood, brain, bone marrow
Results, Tables, Figures
-Revisit the overall order of tables and figures: Table 1 should come before Table 2 (please change the Tables, current Table 1 should move up and also be placed before Fig. 1); tables should always have a headline/title above it; Table 3 belongs to page 7, 8 (2.3.) and not as mentioned in the text;
-Fig. 2 is very good, and should either come as a summary before or after Tables 4, 5
-3.4. Methodology plus Table 5 should also come earlier; please move it up and renumber as Tab 4, (re)use the columns ‘authors’/‘species’/’EE(and add +-stressful events)’/time line of EE” also for Tab 5 (former Tab. 4) – add one column for ‘cytokine’ and one for ‘cell tissue’ (e.g. CD11b etc., what is now stated as “immune cells outcomes’), and split Tab 5 into two (Tab. 5 for ‘brain’ results and Tab. 6 for ‘periphery (blood, bone marrow ..); articles maybe mentioned twice - it should be 6 columns for the new Tab 5 and 6 in summary.
Headline for the discussion part is missing
In the discussion, as for parts of the results, the 16 reviewed articles should be mentioned by the first author names (not just the citation numbers), and discussed by their similarities, differences and overall impact on the development of diseases!
General comments
Please be consistent with abbreviations, use them early and ones in the introduction; please check the spelling of ‘prefrontal cortex’ and always put in small letters, as well as cavity (in Fig. 2); write NK cells (not just natural killers)
Author Response
Abstract, Introduction
- Use ‘articles’ over ‘reports’; ‘influence, role, impact’ instead of ‘performance’, and ‘immune response’; distinguish ‘brain and peripheral tissue’ rather than ‘different tissues’-
R- Dear reviewer, thank you for your consideration, we have adjusted and marked it in yellow.
- 1st citation is not Hebb but a review by Gerd Kempermann, it should be stated as that, Kempermann is also best known for the positive effects ENR has on neuroplasticity.
R- Dear Reviewer, thank you for your observation. Our systematic review used the article by Prof PhD. Kempermann because he refers to the historical content and creation of the first studies with environmental enrichment in rats, developed by prof. Hebb.
- in the Abstract, already write statements: “In the brain, EE ‘positively’ affects. the modulation of cytokines and chemokines … or ‘decreases the release of pro-inflammatory factors’…”
R- Dear reviewer, we have added this part "In the brain, EE positively affected cytokines and chemokines by reducing responses to pro-inflammatory factors in the abstract as requested and it is marked in yellow.
- Please indicate what CD xx, NK cells etc. mean or use broader terms.
R- Dear reviewer, thank you for your contributions. The full name has been added to the abbreviations CD and NK, they are marked in yellow.
- For the Introduction, introduce the various pro, and anti-inflammatory factors first (i.e. like above), briefly state the role of microglia, macrophages, lymphocytes, etc., to make the overall impact of the findings of the 16 articles more relevant; why is only, or specifically, Takai et al. (2019) mentioned in the introduction, and not already a handful of the 16 papers. Why were the criteria for ‘science direct’ suddenly changed, please expand on that (page 4), and since ‘physical exercise’ was a disqualifier (see Fig. 1).
R- Dear reviewer, thank you for your comments. The introduction was revised according to the sequence of information that was requested, initially we described the two types of immunity and their characteristics, to finally arrive at the explanation of pro-inflammatory factors. Takai was cited, because of his results with cancer after exposure to environmental enrichment, which promoted anti-carcinogenic effects in animals and therefore we cite only this study among the 16 included. Science Direct is one of the databases that need to change the search equation since it only receives up to 8 Boolean operators (AND, OR) in its database, so we had to make the adjustments that are described in table1.
- I suggest the following Keywords: inflammation (not immunity), enriched environment (ENR, EE), cytokines, macrophages, blood, brain, bone marrow.
R- Keywords were added as requested and marked in yellow.
Results, Tables, Figures
7- Revisit the overall order of tables and figures: Table 1 should come before Table 2 (please change the Tables, current Table 1 should move up and be placed before Fig. 1); tables should always have a headline/title above it; Table 3 belongs to page 7, 8 (2.3.) and not as mentioned in the text.
R-The tables have been adjusted as per your request dear reviewer, thank you!
8- Fig. 2 is very good, and should either come as a summary before or after Tables 4, 5
R-The tables have been adjusted as per your request dear reviewer, thank you!
9- 3.4. Methodology plus Table 5 should also come earlier; please move it up and renumber as Tab 4, (re)use the columns ‘authors’/‘species’/’EE(and add +-stressful events)’/time line of EE” also for Tab 5 (former Tab. 4) – add one column for ‘cytokine’ and one for ‘cell tissue’ (e.g. CD11b etc., what is now stated as “immune cells outcomes’), and split Tab 5 into two (Tab. 5 for ‘brain’ results and Tab. 6 for ‘periphery (blood, bone marrow ..); articles maybe mentioned twice - it should be 6 columns for the new Tab 5 and 6 in summary.
R-Dear review, we appreciate the considerations and suggestions. We made the suggested changes and renumbered the tables. Now, table 4 is "Sample description and characterization of environmental enrichment protocols and Brain outcomes" and table 5: "Sample description and characterization of environmental enrichment protocols and Cytokines and Immune Cells in Peripheral Tissues”. Also, we made the changes, marked in the text, referring to tables 4 and 5.
10- In the discussion, as for parts of the results, the 16 reviewed articles should be mentioned by the first author names (not just the citation numbers), and discussed by their similarities, differences, and overall impact on the development of diseases!
R- The references were cited as requested, thanks for the comment.
General comments
11- Please be consistent with abbreviations, use them early and ones in the introduction; please check the spelling of ‘prefrontal cortex’ and always put in small letters, as well as cavity (in Fig. 2); write NK cells (not just natural killers)
R-Dear reviewer, the abbreviations have been adjusted and marked in yellow, thank you.
Reviewer 2 Report
The authors conducted a systematic review looking at studies that evaluated inflammatory associated immune factors in rodent brain locations following environmental enrichment.
The authors did a great job describing overall establishing the background for this study and outlining their goals.
Methodologically, the transition from 4417 articles to 1922 is clear (2495 were duplicates). If 1872 of those were excluded based on the inclusion criteria 50 articles would remain. It is unclear why the authors only included 16 in this review.
The way the authors chose to describe the search strategy is difficult to read. This can be omitted or rephrased in the section 2.1 with reference to table 2. I would also suggest a revision to table 2 for similar reasons that includes (for example) Environmental enrichment and enriched environment in one column and the list of immune related terms paired with those in another column.
Figure 1 appears to be copied from a different source. This should be formatted to improve readability and to be consistent with the rest of the article.
What I assume is tables 4 is also very hard to read with variation in font size and text widths making it hard to align outcomes with the authorship information. I also assume that when an equal sign is indicated it meant that outcome was present in EE treated rodents.
The results were not overstated and the discussion is fair given the clearly articulated limitations.
References. Inconsistency is noted in the formatting of references particularly with the use of capital letters in some of the article titles. These should be standardized and meet journal standards.
Minor English editing (mainly for syntax) is recommended.
Author Response
Reviewer 2:
- The authors conducted a systematic review looking at studies that evaluated inflammatory associated immune factors in rodent brain locations following environmental enrichment. The authors did a great job describing overall establishing the background for this study and outlining their goals. R- Dear reviewer, thank you for the excellent description of our work!
- Methodologically, the transition from 4417 articles to 1922 is clear (2495 were duplicates). If 1872 of those were excluded based on the inclusion criteria 50 articles would remain. It is unclear why the authors only included 16 in this review.
R- Dear reviewer, thank you for the excellent comment. After the screening process of the articles carried out by two independent authors, 50 were left for full-text reading and after that we observed that 34 did not meet the eligibility criteria, leaving 16 articles for inclusion.
- The way the authors chose to describe the search strategy is difficult to read. This can be omitted or rephrased in section 2.1 with reference to table 2. I would also suggest a revision to table 2 for similar reasons that include (for example) Environmental enrichment and enriched environment in one column and the list of immune-related terms paired with those in another column.
R- Dear reviewer, to simplify the understanding of the point of the search equation, we decided to just cite Table 1, where all the search equations used by the database are fully described. Thank you for your comment.
- Figure 1 appears to be copied from a different source. This should be formatted to improve readability and to be consistent with the rest of the article.
R- Dear reviewer, thank you for the excellent observation, Figure 1 is the model flowchart provided by the PRISMA platform, which provides several models for filling, following the protocol of the systematic review, we followed this model and added it to our review respecting all the criteria. The figure is revised and added in the corresponding section.
- What I assume is tables 4 is also very hard to read with variation in font size and text widths making it hard to align outcomes with the authorship information. I also assume that when an equal sign is indicated it meant that outcome was present in EE treated rodents.
R- Dear reviewer, tables have been updated and standardized in terms of font size and view. The equal sign was added to the legend of Table 5, which deals with the main results, it is about not having statistically significant differences in environmental enrichment between the control or comparator group.
- The results were not overstated, and the discussion is fair given the clearly articulated limitations.
R- Dear reviewer, thank you for the excellent description of this section.
- References. Inconsistency is noted in the formatting of references, particularly with the use of capital letters in some of the article titles. These should be standardized and meet journal standards.
R- Dear reviewer, the references have been adjusted following the journal's standard and the titles are all in lowercase letters. Thank you for your consideration.
- Minor English editing (mainly for syntax) is recommended.
R- The work underwent a review in English by native speakers.
Round 2
Reviewer 1 Report
The authors have adequately addressed my comments. Yet, a few remain:
Abstract: “In the brain, EE diminishes the risk of inflammation and thus the release of pro-infl. cytokines and chemokines was reduced.” ?!
“Natural killer cells” can be abbrev. after the first mention (please have another look concerning its spelling throughout); CD11b, CD27, CD19 can be as this (no spelling out is necessary here, in the abstract)
“… role in preventing numerous diseases that (have) require? a pro-inflammatory response and immunosuppressive condition.”
Cit. 1 is still not mentioned in the sentence; one would expect a Hebb citation not Kempermann; please add for example “… postulated by Hebb in 1940 … and further studied in adult neurogenesis.”
Author Response
Dear Review,
Systematic Review. Cytokines and immune cells profile in different tissues of rodents induced by Environmental Enrichment
1. Abstract: “In the brain, EE diminishes the risk of inflammation and thus the release of pro-infl. cytokines and chemokines were reduced.” ?!
R- Dear reviewer, thank you observation, in the abstract, we added the text making it clearer that environmental enrichment reduced the production of pro-inflammatory cytokines and chemokines in the brain. It is marked in the text.
2.“Natural killer cells” can be abbrev. after the first mention (please have another look concerning its spelling throughout); CD11b, CD27, CD19 can be as this (no spelling out is necessary here, in the abstract)
˜R- Dear reviewer, thank you for the comment, in the abstract We have added the acronyms as requested.
3.“… role in preventing numerous diseases that (have) require? a pro-inflammatory response and immunosuppressive condition.”
R- Dear reviewer, in the conclusion of the abstract we made it clearer that environmental enrichment can act as a non-pharmacological tool in chronic diseases that have inflammatory and immunosuppression as part of their pathophysiology. Thanks for the excellent comment.
4.cit. 1 is still not mentioned in the sentence; one would expect a Hebb citation not Kempermann; please add for example “… postulated by Hebb in 1940 … and studied further in adult neurogenesis.”
R- Dear reviewer, the original quote from Hebb was added to the manuscript by removing the quote from Kempermann, yet we added more details about his postulate. It's all marked in the intro.